# Machine Learning-Based Screening Tool for Lung Adenocarcinoma Via Gut Microbiome Analysis

Jeong Kyu Lee
*Computer Science Program*
*New York University Abu Dhabi*
Abu Dhabi, United Arab Emirates
jkl499@nyu.edu

Mai Oudah
*Computer Science Program*
*New York University Abu Dhabi*
Abu Dhabi, United Arab Emirates
mai.oudah@nyu.edu

*Abstract*—Lung adenocarcinoma (LUAD) represents a major global health challenge requiring more accessible and non-invasive screening methods. Traditional diagnostic approaches such as computed tomography or biopsies are effective but costly, resource-intensive, and carry associated risks. This study leverages gut microbiome data and machine learning techniques to develop a non-invasive pre-screening tool for LUAD. Using a dataset of 107 fecal samples (43 LUAD and 64 healthy controls), we explored the performance of nine machine learning algorithms and four distinct feature sets generated through feature selection methods to identify informative microbial biomarkers and construct accurate classification models. Our results demonstrate that feature selection significantly enhances model performance compared to baseline approaches. A Random Forest model combined with Correlation-based Feature Selection achieved an Area Under the Curve of 0.9967. Key taxa including *Prevotella*, *Coprococcus*, *Phascolarctobacterium*, *Bilophila*, *Blautia*, *Enterococcus*, and *Bacteroides* emerged as potential biomarkers. Functional predictions using PICRUSt2 revealed significant alterations in folate metabolism, methylation cycles, and photosynthetic bacterial activity, highlighting disrupted gut microbiome function in LUAD patients. These findings align with previous studies and suggest promising directions for non-invasive and cost-effective screening methods.

*Index Terms*—Biomarkers, classification algorithms, feature selection, gut microbiome, metabolic pathways, lung adenocarcinoma, machine learning, medical screening

## I. INTRODUCTION

Lung cancer represents the leading cause of cancer-related deaths globally, posing a significant public health challenge. According to the International Agency for Research on Cancer, lung cancer accounted for 1.8 million deaths in 2020, representing 18% of all cancer fatalities [1]. Among different forms of lung cancer, lung adenocarcinoma (LUAD) is the most common subtype [8]. This statistic underscores the urgent need for more effective and accessible treatment strategies and screening methods to enhance early detection.

Traditional diagnostic methods, such as computed tomography (CT) scans, Positron Emission Tomography (PET) scan, bronchoscopy and biopsy, present significant barriers to widespread screening implementation. These approaches are not only costly and invasive but also involve risks such as radiation exposure, which can lead to additional health complications [6]. The potential for false positives and negatives in current screening tools creates additional challenges,

potentially leading to unnecessary interventions and increased healthcare costs [6]. Furthermore, these barriers contribute to healthcare disparities, leaving many populations underserved, particularly in low-resource settings and geographically isolated areas [8].

The human microbiome field offers promising opportunities for developing non-invasive pre-screening tools [26]–[34]. The microbiome represents a complex ecosystem of microorganisms residing throughout the human body, with composition and function significantly altered in various disease states, including cancer [9]. Specific changes in the microbiome may serve as reliable indicators for determining when traditional LUAD screening methods should be pursued. Here, we aim to investigate the potential for a cheaper non-invasive LUAD screening tool when Stool samples, i.e. Gut Microbiome, are utilized as the non-invasive material, while a set of identified informative biomarkers may lead to a cheeper targeted sequencing approach.

Machine learning algorithms excel in extracting patterns from large and complex datasets, making them ideal for analyzing microbiome data to detect subtle alterations associated with LUAD [5]. This research addresses the central question: "What is the optimal combination of machine learning algorithm and feature selection method for developing a gut microbiome-based screening tool for LUAD?"

By leveraging these computational tools and the non-invasive nature of microbiome sampling, this research aims to provide a reliable pre-screening tool that can guide decisions about pursuing more invasive diagnostic procedures. This approach offers particular benefits for low-resource settings, developing countries, and geographically isolated areas, potentially reducing global healthcare inequity and enhancing public health outcomes.

## II. RELATED WORK

Previous studies have investigated the relationship between microbiomes and lung cancer using both taxonomic diversity analysis and machine learning approaches. Liu et al. [2] explored gut fungal profiles as non-invasive biomarkers for early-stage LUAD detection using 299 fecal samples (181 LUAD patients and 118 healthy controls). Their Random Forest models achieved excellent Area Under the Curve (AUC)

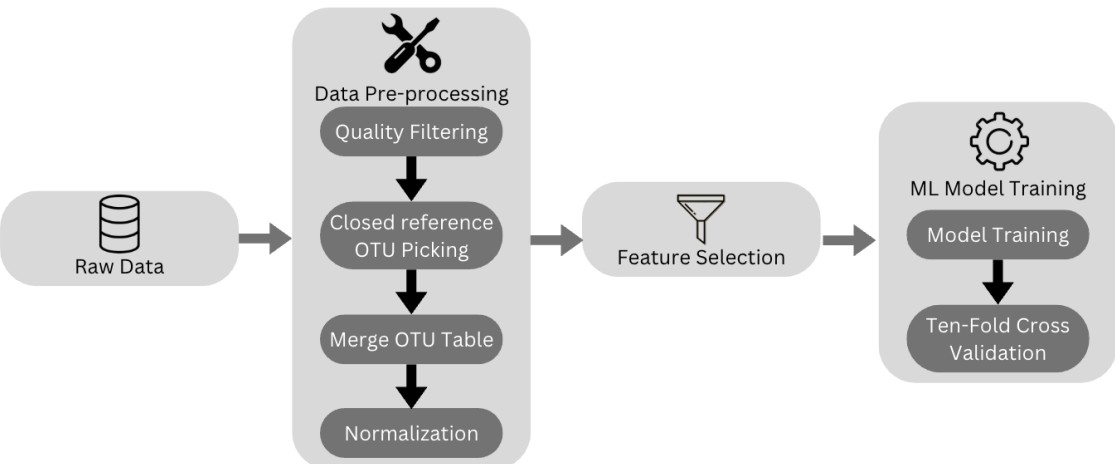

Fig. 1. Overall Pipeline for Classification Model Development

values ranging from 0.88-0.96 across multiple validation cohorts, identifying significant biomarkers including decreased *Candida* and increased *Saccharomyces*, *Aspergillus*, and *Apiotrichum* in LUAD patients.

Guo et al. [10] examined relationships between gut and lung microbiomes in LUAD patients using paired fecal and bronchoalveolar lavage fluid samples from 42 patients. They identified reduced $\alpha$-diversity and significantly altered $\beta$-diversity in gut microbiome of LUAD patients compared to healthy controls, finding increased abundances of several bacteria including *Flavonifractor*, *Eggerthella*, and *Clostridium*. The study revealed significant alterations in functional pathways involving leucine, propanoate, and fatty acids associated with LUAD progression.

Ni et al. [3] combined gut microbiome and serum metabolomics to study early-stage non-small cell lung cancer, identifying 26 genera and 123 metabolites significantly altered in early-stage patients. Their findings suggested that dysregulation of sphingolipid metabolism pathways could represent potential therapeutic strategies.

Liang et al. [4] applied various machine learning approaches including Random Forest, Support Vector Machines, and Neural Networks to predict immunotherapy response using gut microbiome data from 128 patient samples, achieving AUC values between 0.67-0.75. Freitas et al. [7] utilized Random Forest to classify five different cancer types using microbiome data, achieving accuracy ranging between 72-96%.

## III. METHODOLOGY

The methodology follows a structured pipeline encompassing data processing, feature extraction (which generates the microbial composition in terms of operational taxonomic units (OTUs)), feature selection, machine learning model development, and validation, as illustrated in Figure 1.

### A. Dataset

The dataset we analyze in this work is publicly available on the National Center for Biotechnology Information (NCBI) Repository under accession number PRJNA906201 [10], [15]. It was collected and posted by the Institute of Microbiology, Chinese Academy of Sciences. The dataset consists of samples from 43 LUAD patients and 64 healthy controls, providing 107 fecal samples processed using 16S rRNA amplicon sequencing [10].

### B. Feature Extraction

Feature extraction involved several preprocessing steps to prepare the feature space from raw data. First, extracted 16S rRNA sequences were trimmed based on quality criteria including a quality score threshold of 30, minimum length of 80 bases, and quality score offset of 33. An operational taxonomic unit (OTU) table was generated through closed-reference OTU picking via QIIME2 [12] using 97% sequence similarity threshold against the Greengenes 13.8 reference database [11]. Abundances were normalized to obtain relative abundances and scaled by a factor of 1,000,000 for improved readability. This process resulted in a 107 × 10,735 table including sample labels.

### C. Feature Selection

Feature selection methods address dimensionality reduction challenges inherent in microbiome data, where small sample sizes relative to large feature spaces can lead to overfitting [14], [26]. Four different feature sets were evaluated: one baseline set using all features, and three sets generated through feature selection methods implemented via python-weka-wrapper3 [13]. We examine the performance of the models trained on the set of informative features each feature selection method generates separately.

Correlation-based Feature Subset Selection (CFS), which identifies a subset of features that are highly correlated with the target variable (class) but have low inter-correlation among

TABLE I
PERFORMANCE COMPARISON OF FEATURE SELECTION METHODS ACROSS MODELS

| | Baseline | | | | CFS | | | | InfoGain | | | | GainRatio | | | |
|---|---|---|---|---|---|---|---|---|---|---|---|---|---|---|---|---|
| | P | R | F | AUC | P | R | F | AUC | P | R | F | AUC | P | R | F | AUC |
| NB | .5747 | .5773 | .5551 | .5436 | .8930 | .8700 | .8661 | .9345 | .6510 | .6327 | .6183 | .6257 | .6842 | .6191 | .6058 | .6576 |
| NBM | .5870 | .5927 | .5750 | .5735 | .7241 | .7082 | .7014 | .7076 | .6886 | .6718 | .6632 | .6647 | .6724 | .6545 | .6433 | .6303 |
| LR | .6641 | .6536 | .6448 | .6498 | .9138 | .8982 | .8948 | .9137 | – | .6527 | – | .6963 | – | .6655 | – | .7130 |
| J48 | .7025 | .6809 | .6731 | .6255 | .8362 | .8218 | .8170 | .8061 | .8362 | .8218 | .8170 | .8061 | .7725 | .7364 | .7343 | .7742 |
| DT | .7267 | .7000 | .6884 | .7730 | .7047 | .6736 | .6658 | .7340 | – | .7382 | – | .8461 | .7213 | .7109 | .6994 | .7558 |
| SMO | .6938 | .6627 | .6434 | .6296 | .9240 | .9073 | .9036 | .9012 | .6149 | .5855 | .5783 | .5977 | .6849 | .5927 | .5813 | .6249 |
| RF | – | .6527 | – | .7961 | **.9593** | **.9527** | **.9514** | **.9967** | .8364 | .8218 | .8160 | .8685 | .8188 | .8055 | .7985 | .8917 |
| AB | .7632 | .7264 | .7108 | .8633 | .8572 | .8309 | .8277 | .9639 | .8120 | .7927 | .7895 | .8654 | .8278 | .8136 | .8120 | .8492 |
| ST | – | .5982 | – | .5000 | .9392 | .9245 | .9236 | .9838 | .8262 | .8036 | .8007 | .8802 | .8267 | .8145 | .8152 | .9008 |

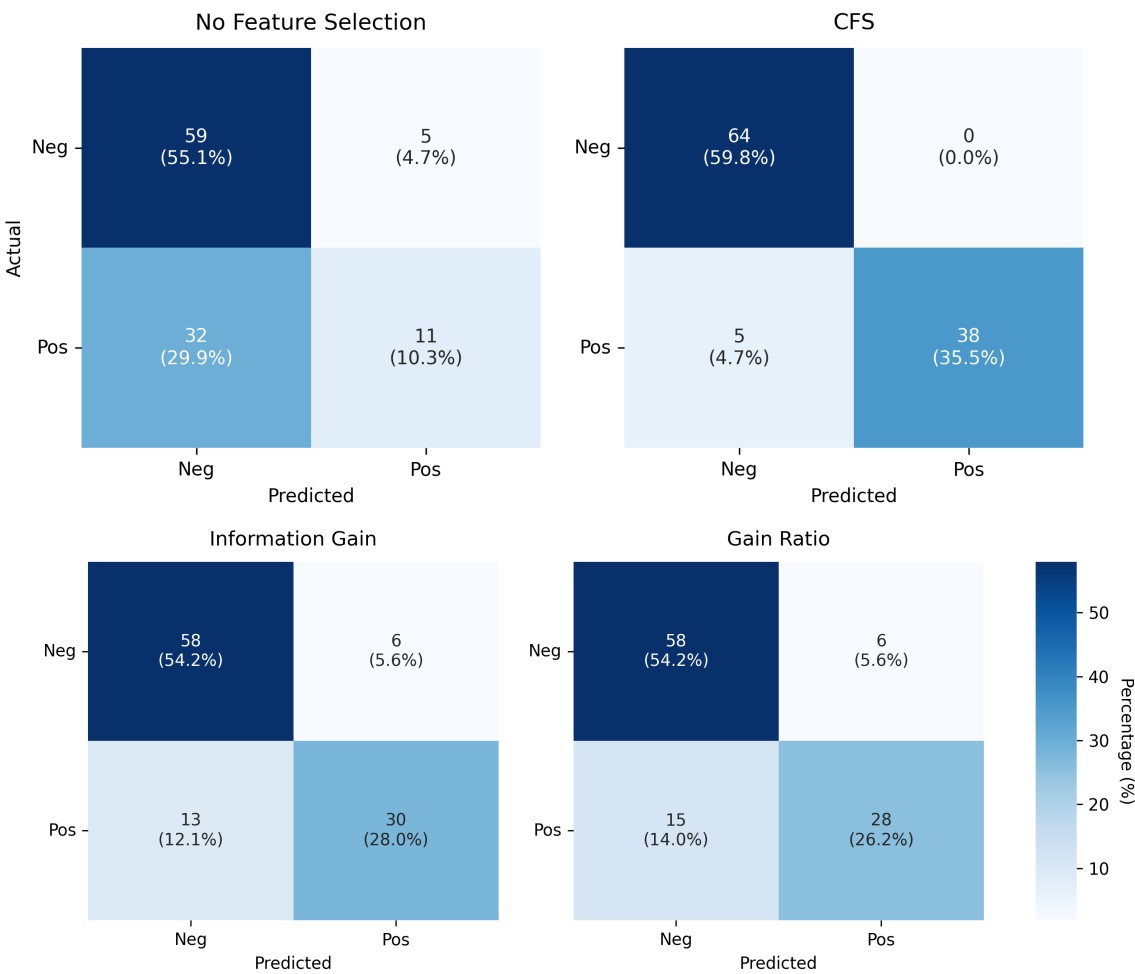

Fig. 2. Confusion matrices of Random Forest model across different feature selection methods

themselves, using default parameters selected an average of 75 features across runs. Information Gain (IG), which evaluates the worth of a feature by measuring the amount of information gained about the targeted class when considering that feature, was configured to select the top 55 features, while Gain Ratio, which normalizes the Information Gain score by considering the intrinsic information of a feature as well, selected the top 205 features. Feature counts for IG and GainRatio were determined experimentally by evaluating various values across

nine machine learning models to identify optimal numbers yielding highest AUC values.

### D. Model Training and Validation

Nine machine learning models were implemented via python-weka-wrapper3 [13]: Naive Bayes (NB), Naive Bayes Multinomial (NBM), Logistic Regression (LR), Decision Tree (J48), Decision Table (DT), Sequential Minimal Optimization (SMO), Random Forest (RF), Adaptive Boosting (AB), and

Stacking (ST). The nine machine learning algorithms were chosen based on their performance in previous microbiome studies.

Hyperparameters were optimized using GridSearch for optimal results and evaluated through 10-fold cross-validation. The training dataset is split into 10 partitions and in every iteration 9 partitions are used for training and the remaining one is used for testing. In every iteration, a feature selection method is applied to the 9 partitions used for training in order to identify the informative features given that particular training set for model training (the testing partition is then represented via the selected informative features and used to evaluate the trained model). Any reported AUC is averaged over the 10 iterations. The results of Neural networks were not reported due to the poor performance compared to the other 9 methods. This is mostly due to the small size of the training dataset, which favors classical machine learning approaches [16].

*E. Functional Pathway Analysis*

Functional pathway analysis was performed using PICRUSt2 (v2.5.2) to infer microbial metabolic pathways from 16S rRNA data [39]. Sequences were placed into reference phylogeny using EPA-NG, followed by hidden-state prediction of gene families via castor. Metagenome predictions were normalized by 16S copy number, and pathway abundances were inferred using MinPath. Differential abundance testing was performed on pathways identified by CFS, Gain Ratio, and IG methods.

## IV. Experimental Results

The experiment revealed comprehensive performance metrics across various machine learning models and feature selection approaches for LUAD classification using gut microbiome data. Table I presents detailed comparisons showing Precision (P), Recall (R), F-score (F), and AUC metrics for all combinations, with primary focus on AUC as it provides comprehensive assessment of classification ability across all thresholds [17].

Without feature selection, baseline models showed moderate but varied performance, with Adaptive Boosting achieving the highest baseline AUC of 0.8633. The CFS feature selection method demonstrated superior performance across nearly all models. The Random Forest with CFS combination achieved optimal performance with AUC of 0.9967, precision of 0.9593, recall of 0.9527, and F-measure of 0.9514, representing the highest values across all metrics and combinations. Stacking with CFS followed closely with AUC of 0.9838.

Other feature selection methods showed moderate improvements over baseline. Information Gain demonstrated reasonable performance enhancement, with Stacking achieving the highest AUC of 0.8802. GainRatio also showed improvements, with Stacking reaching AUC of 0.9008.

Figure 2 displays confusion matrices for Random Forest across different feature spaces, illustrating improvements in classification performance. The CFS method achieved minimum misclassifications, demonstrating highest accuracy, while

IG and GainRatio methods also showed improved performance over baseline.

Analysis of significant taxa identified by each feature selection method revealed important biomarkers. Figure 3 presents $\log_2$ fold changes for identified biomarkers across all methods, with biomarkers listed in order of significance.

The CFS method revealed several key microbial signatures with high statistical significance. Notable enrichment was observed in multiple *Bilophila* species with consistent positive fold changes (0.52-0.62), and *[Paraprevotellaceae]* showing the highest $\log_2$ fold change of 11.18. Significant alterations in *Lachnospiraceae* family members, particularly *Coprococcus* with significant decrease (fold change -1.388), were identified. Multiple *Phascolarctobacterium* family members emerged as key biomarkers, along with enriched taxa including *Streptococcus*, *Prevotella*, *Oscillospira*, *Dorea*, *Bacteroides*, and *Bacteroidaceae*.

All three methods consistently identified *Erysipelotrichaceae* as a key biomarker, with *Lachnospiraceae* family showing significant alterations across all methods, and *Enterococcus* species selected in IG and Gain Ratio methods.

Pathway analysis revealed significant dysregulation of microbial metabolic processes in LUAD. Figure 4 presents the most significantly altered MetaCyc pathways across all feature selection methods, ranked by absolute t-test values.

Across all three methods, several functional pathways consistently emerged as significantly altered. N10-formyl-tetrahydrofolate biosynthesis (1CMET2-PWY) and S-adenosyl-L-methionine cycle I (PWY-6151) were consistently downregulated, while adenosine and guanosine deoxyribonucleotide biosynthesis pathways (PWY-7220 and PWY-7222) were upregulated. Chlorophyllide a biosynthesis pathways (CHLOROPHYLL-SYN, PWY-5531, and PWY-7159) involved in bacterial photosynthesis were notably downregulated in InfoGain and GainRatio results.

These findings point to major disruption in methylation-related and nucleotide biosynthesis pathways in LUAD-associated microbiome. The downregulation of folate metabolism and SAM cycle suggests impaired one-carbon metabolism affecting DNA synthesis and methylation [25], [38]. Simultaneous upregulation of de novo dNTP biosynthesis pathways implies potential compensatory activity for increased microbial DNA replication in LUAD environments [35], [36].

## V. Discussion

Our findings align with and extend previous research using the same underlying dataset as Guo et al. [10]. Several bacterial taxa emerged as common biomarkers including *Coprococcus*, *Dorea*, *Lachnospiraceae* family members, *Ruminococcus*, *Prevotella*, *Enterococcus*, *Fusobacterium*, and *Bacteroides*. Our analysis highlighted additional genera including *Blautia*, *Bifidobacterium*, *Phascolarctobacterium*, and *Erysipelotrichaceae* not emphasized in previous work.

These discrepancies are expected given methodological differences, as our study focuses on machine learning approaches

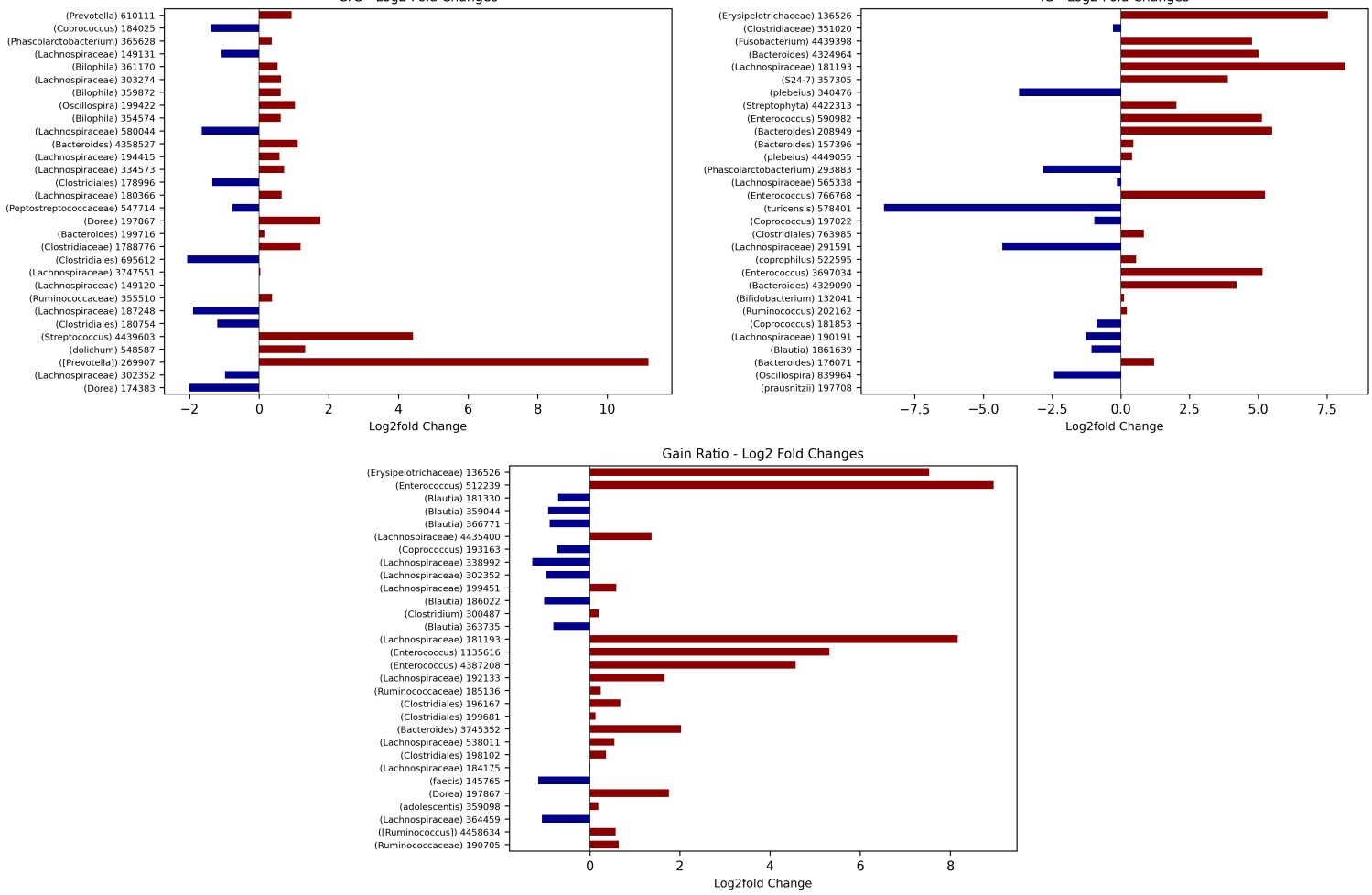

Fig. 3. Top 30 most significant taxa of each feature selection method. Red bars: +ve log$_2$ fold change; Blue bars: -ve log$_2$ fold change.

using feature selection methods on gut microbiome data alone, while Guo et al. performed cross-site analysis of both lung and gut microbiomes. The core similarities confirm and reinforce key microbiota identified in previous research.

Our models achieved strong performance metrics, with Random Forest using CFS feature selection reaching AUC of 0.9967 and Stacking with CFS achieving AUC of 0.9838. Other studies using different datasets and methodologies have reported AUC values ranging from 0.67-0.98 [2], [4], [7]. While direct performance comparisons cannot be made due to methodological differences, our analysis represents the first machine learning application to this specific dataset.

The identified biomarkers including *Prevotella*, *Coprococcus*, *Phascolarctobacterium*, *Bilophila*, *Blautia*, *Enterococcus*, *Fusobacterium*, and *Bacteroides* align with biomarkers reported in literature, confirming their importance as key genera [3], [4], [7]. In practice, we envision that the top-ranked taxa will be used as a cost-effective biomarker panel. With the reduced set of features (to only the top informative biomarkers) for targeted sequencing, we hope to provide a cheaper and faster method for generating relative abundances required by

the classification model in order to predict a class (normal vs. cancer) for a previously unseen sample.

Functional pathway analysis highlighted consistent metabolic disruptions in LUAD microbiome, including alterations in N10-formyl-tetrahydrofolate biosynthesis, S-adenosyl-L-methionine cycle, and de novo nucleotide biosynthesis. These patterns suggest restructuring of both microbial composition and metabolic roles, reinforcing the possibility that LUAD accompanies such significant changes.

The dataset originates entirely from a Chinese population, and microbiome compositions may be influenced by genetic, environmental and dietary factors specific to this population. Geographic and ethnic differences can affect gut microbial profiles [20]. While identified biomarkers demonstrate strong predictive potential within the Chinese cohort, validation with diverse datasets is crucial for global applicability.

## VI. CONCLUSION

This study investigated gut microbiome data and machine learning techniques for non-invasive LUAD screening, achieving strong classification performance through systematic eval-

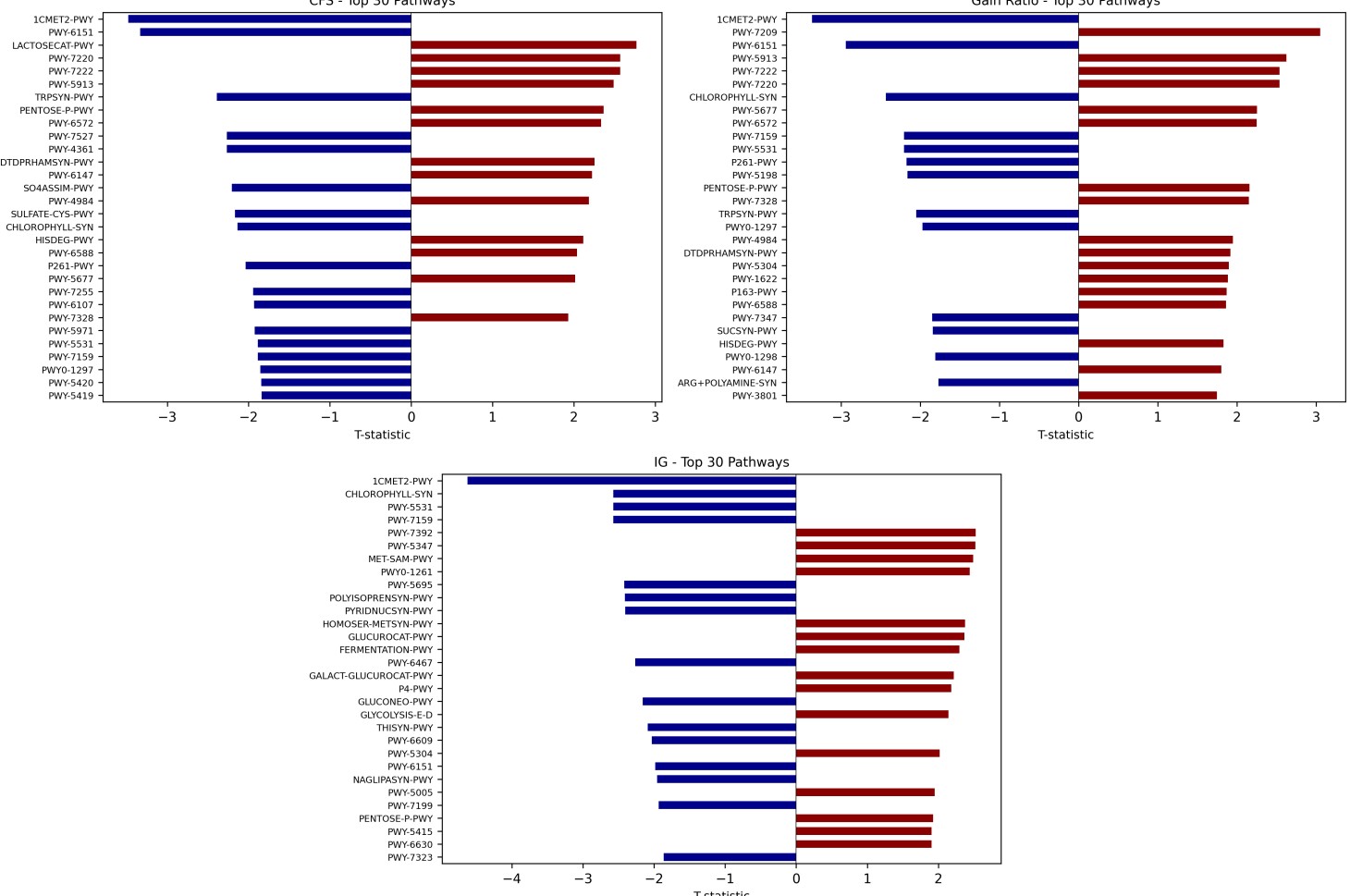

Fig. 4. Top 30 most significantly altered MetaCyc pathways of each feature selection method. Red bars: +ve log₂ fold change; Blue bars: -ve log₂ fold change.

uation of nine machine learning models and multiple feature selection methods. The Random Forest with CFS feature selection combination yielded optimal performance with AUC of 0.9967, demonstrating the significance of feature selection in improving classification outcomes and identifying informative microbial biomarkers.

Key taxa including *Prevotella*, *Coprococcus*, *Phascolarctobacterium*, *Bilophila*, *Blautia*, *Enterococcus*, and *Bacteroides* emerged as potential biomarkers. Functional pathway analysis revealed consistent alterations in folate metabolism, methylation pathways, and nucleotide biosynthesis, supporting the presence of systemic microbial shifts in LUAD.

The implications for global healthcare are significant, particularly for low-resource settings. 16S rDNA sequencing costs approximately $47.91 per sample [18], significantly lower than CT scans at $1,565 per procedure [19]. Our machine learning approach with feature selection reduces required biomarkers, further lowering costs while providing reliable, non-invasive pre-screening capabilities.

As Future work, we plan to validate our approach on independent and larger datasets of diverse cohorts. Moreover, we plan to explore additional functional aspects, such as gene expression, and ensemble methods with diverse base learners. Furthermore, examining two feature selection methods in parallel is worth investigating.

## ACKNOWLEDGMENT

The research was conducted using the High-Performance Computing (HPC) resources provided by New York University Abu Dhabi.

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
