# OpenReview forum: "Machine Learning-Based Screening Tool for Lung Adenocarcinoma Via Gut Microbiome Analysis"
_IEEE.org/EMBS/BHI/2025/Conference — BHI 2025_

### Official Review · Reviewer_omKH · 2025-06-25
**Machine Learning-Based Screening Tool for Lung Adenocarcinoma Via Gut Microbiome Analysis**

**Confidence:** 4
**Clarity Of Writing:** good
**Clinical Significance:** great
**Methodological Novelty:** fair
**Overall Rating:** 7

**Experiments And Results:**

good

**Questions For The Authors:**

Have you validated your findings on independent cohort from different ethnic populations? Single Chinese population is major limitation for clinical tool. (Critical for generalizability assessment)
Can you explain biological plausibility of achieving 0.9967 AUC with only 107 samples? This performance seems unrealistic without overfitting. (Essential for credibility)
Where is ethical approval statement for human sample collection? This is required information for biomedical research.

**Strengths:**

Clear clinical motivation: The paper is addressing very important clinical need for non-invasive and cost-effective screening methods for lung adenocarcinoma. Authors are providing strong justification through cost comparison ($47.91 for 16S rRNA sequencing versus $1,565 for CT scan procedures).
Comprehensive methodology: The study is evaluating nine different machine learning algorithms with multiple feature selection methods, which provides thorough comparison of approaches.
Exceptional performance metrics: Random Forest with CFS combination achieved very high performance with AUC of 0.9967, precision of 0.9593, recall of 0.9527, and F-measure of 0.9514.
Functional pathway analysis inclusion: Using of PICRUSt2 analysis for examining metabolic pathway alterations is adding biological insight beyond simple taxonomic classification.
Potential for low-resource settings: The approach is offering particular benefits for developing countries and geographically isolated areas.

**Summary Of The Paper:**

This manuscript is presenting machine learning-based methodology for development of non-invasive screening tool for lung adenocarcinoma (LUAD) by using gut microbiome data. The authors are utilizing dataset which contains 107 fecal samples (43 samples from LUAD patients and 64 samples from healthy control subjects) for exploring nine different machine learning algorithms in combination with four distinct feature selection methods. Main objective of study is identification of informative microbial biomarkers and construction of accurate classification models. Best performance was achieved by Random Forest model when combined with Correlation-based Feature Selection (CFS), which reached Area Under the Curve (AUC) of 0.9967. The paper is identifying several key taxa as potential biomarkers, including Prevotella, Coprococcus, Phascolarctobacterium, Bilophila, Blautia, Enterococcus, and Bacteroides. Additionally, functional predictions by using PICRUSt2 revealed significant alterations in folate metabolism, methylation cycles, and photosynthetic bacterial activity in patients with LUAD.

**Weaknesses:**

Very limited sample size: Only 107 samples (43 LUAD, 64 controls) from single Chinese population is severely limiting generalizability of findings.
Absence of independent validation: All results are based on cross-validation without independent test set, which may be leading to overoptimistic estimates.
Statistical rigor is lacking: No p-values, confidence intervals, or significance tests are provided for performance metrics.
Overfitting concerns: The extremely high AUC of 0.9967 with small dataset is raising serious concerns about overfitting.
Limited biological interpretation: Paper provides minimal discussion about biological mechanisms linking identified microbes to lung cancer.
The paper has solid scientific merit, addresses an important problem with great clinical significance, and presents good experimental work. The weaknesses I identified (validation issues) are why it's "weak accept" rather than "accept" or "strong accept," but it's definitely above the borderline threshold.

---

### Official Review · Reviewer_RiCw · 2025-06-25
**Review on Machine Learning-Based Screening Tool for Lung Adenocarcinoma Via Gut Microbiome Analysis**

**Confidence:** 4
**Clarity Of Writing:** fair
**Clinical Significance:** good
**Methodological Novelty:** poor
**Overall Rating:** 2
**Final Rating:** 3

**Experiments And Results:**

fair

**Questions For The Authors:**

Please see the weaknesses section.

**Strengths:**

1. The application of ML in gut microbiome for cancer diagnosis, let alone LUAD screening, is a very interesting field with many recent high-quality publications.
2. Comparison of several methods/architectures.

**Summary Of The Paper:**

The paper presents a comparison amongst several ML models and feature selection techniques to classify LUAD patients vs healthy controls based on gut microbiome 16s rRNA data. Feature selection is clearly needed due to the large feature space with limited number of samples would otherwise result in overfitting models. Experimental results validate this hypothesis.

**Weaknesses:**

1. It's not clear whether multiple feature selection methods were employed in parallel. Using CFS with another feature selection method would might increase performance even further.
2. The rationale behind the association between gut microbiome and LUAD is not discussed/elaborated enough.
3. Small (narrow) neural networks could have still been employed, despite the small dataset size.
4. No technical novelty is presented in this paper.
5. Feature selection methods were not discussed/elaborated with respect to their technical details.
6. The clinical and biological explanation of the results is inadequate. I would have wanted to read a more detailed interpretation of the results, with specific discussion on how future analysis could shed light into the disease, prevention and/or treatment options. Associations with other comorbidities/covariates (e.g. BMI, smoking, etc.) is also necessary to conclude that the model is identifying key features directly linking microbiome and LUAD and does not pick up noise from other covariates.

---

### Official Review · Reviewer_PD7q · 2025-07-01
**Promising but Overfitted? A Critical Review of Gut Microbiome-Based LUAD Screening Using Machine Learning**

**Confidence:** 4
**Clarity Of Writing:** good
**Clinical Significance:** great
**Methodological Novelty:** good
**Overall Rating:** 6
**Final Rating:** 7

**Experiments And Results:**

good

**Questions For The Authors:**

1. What validation strategy was used to prevent overfitting?
With such high AUC values and a small dataset, how did you ensure the model isn’t overfitted?
2. Why were these nine models chosen, and why were neural networks excluded?
A brief rationale for model selection would improve transparency.
3. How do you envision this model being applied in practice?
Would it be a standalone classifier or would it guide further diagnostic testing (e.g., CT scan referral)?
4. Can the key taxa be distilled into a simplified diagnostic panel?
Is a reduced set of biomarkers feasible for targeted qPCR or similar methods?
5. Have you considered validating your approach on independent datasets?
If not, do you plan to do so in the future?

**Strengths:**

1. Clear Clinical Motivation: The author clearly establishes the need for non-invasive, cost-effective LUAD screening tools, especially in low-resource settings. The burden of LUAD is effectively contextualized within the global health landscape.
2. Relevant and Representative Dataset: The authors use a publicly available, high-quality dataset (PRJNA906201) from a prior study, ensuring reproducibility and relevance to LUAD screening.
3.  Literature Contextualization: The related work section provides a well-organized summary of prior studies using both taxonomic and functional microbiome analysis, including other machine learning applications.
4.  Alignment with Prior Work: The discussion appropriately compares the authors’ findings with those of Guo et al., who used the same dataset. Similarities and methodological differences are clearly explained, supporting consistency with previous literature.
5. Acknowledgment of Population Bias: The authors recognize that the dataset is based solely on Chinese patients and discuss potential limitations due to genetic and environmental variation — a key consideration for generalizability.
6. Functional Insight via Pathway Analysis: Beyond classification, the PICRUSt2-based analysis adds interpretability by linking LUAD to disrupted microbial functions in one-carbon metabolism and DNA synthesis.

**Summary Of The Paper:**

This paper proposes a non-invasive pre-screening approach for lung adenocarcinoma (LUAD) using gut microbiome data and classical machine learning techniques. The authors evaluate nine machine learning models across four feature sets, using a publicly available 16S rRNA dataset containing 107 fecal samples (43 LUAD and 64 controls). Feature selection methods (CFS, Information Gain, GainRatio) are shown to significantly improve performance, with Random Forest + CFS achieving an AUC of 0.9967. Functional pathway analysis using PICRUSt2 reveals disruptions in folate metabolism, nucleotide biosynthesis, and methylation pathways. The study supports the viability of gut microbiome signatures as biomarkers for LUAD and highlights the potential of machine learning in developing low-cost screening tools.

**Weaknesses:**

1. Potential Overfitting and Unrealistic Performance
The reported AUC of 0.9967 is extremely high for such a small sample size with high-dimensional microbiome features. The authors do not report whether they used nested cross-validation, permutation testing, or external validation — standard methods to mitigate overfitting. Without this, the results may reflect model overfit rather than true generalization.
2. No Rationale for Model Selection
The paper includes nine machine learning models but does not explain why these were chosen. Were they based on prior performance in microbiome studies? Why were neural networks excluded entirely, even lightweight architectures?
3. Limited Discussion on Clinical Translation
While the classification models perform well, it is unclear how these would be deployed in practice. Could the top-ranked taxa be converted into a cost-effective biomarker panel? How many microbial features are realistically required?
4. Unclear Reference to “Traditional Diagnostic Methods”
The introduction criticizes traditional LUAD diagnostics as costly and invasive but does not specify what these are (e.g., CT scans, PET, bronchoscopy, biopsy). Providing this context would help general readers and justify the motivation.
5. Visual and Formatting Issues
5.1 Figure 1: Mentions "OTU" without defining it in the caption; acronyms should be expanded for clarity.
5.2 Figure 2: Labels and legends are small; readability should be improved.
5.3 Table 1: Random Forest results are bolded, but in such a large table, additional emphasis (e.g., bold + underline or color) would improve visibility.

---

### Official Review · Reviewer_Lfyj · 2025-07-17
**A journal quality study**

**Confidence:** 3
**Clarity Of Writing:** great
**Clinical Significance:** great
**Methodological Novelty:** great
**Overall Rating:** 7
**Final Rating:** 8

**Experiments And Results:**

great

**Questions For The Authors:**

I have no questions.

**Strengths:**

1. The language used is clear and concise.
2. The study is supported with sufficient tables and figures.
3. References used are timely, highlighting the relevance of the paper.
4. Comparisons done with the existing literature highlights the impact of the paper well.

**Summary Of The Paper:**

In this study, authors used gut microbiome data data and machine learning (ML) methods to develop a non-invasive pre-screening tool for lung adenocarcinoma (LUAD). Authors used a dataset of 107 fecal samples which consists of patients and controls which they explored the performance of 9 ML algorithms. Results indicate that when feature selection is used with the ML models it significantly improves the model performance compared to regular approaches. Authors conclude that the findings align with previous studies and suggest promising directions for non-invasive and cost effective screening methods.

**Weaknesses:**

1. Authors are recommended to check the conference template and address any issues as well as typographic errors.
2. Figures should be displayed and explained better, currently they are too small and they don't contain a legend for the blue and red colors.